# Law of Vision Representation in MLLMs

**Shijia Yang**[1]     **Bohan Zhai**[4]     **Quanzeng You**[4]

**Jianbo Yuan**[4]     **Hongxia Yang**[3]     **Chenfeng Xu**[2*]

[1]STANFORD UNIVERSITY [2]UC BERKELEY
[3]THE HONG KONG POLYTECHNIC UNIVERSITY [4]INDEPENDENT RESEARCHER

## Abstract

We introduce the "Law of Vision Representation" in multimodal large language models (MLLMs), revealing a strong correlation among cross-modal alignment, vision representation correspondence, and overall model performance. We quantify these factors using the cross-modal **A**lignment and **C**orrespondence scores. Extensive experiments across fifteen distinct vision representation settings and evaluations on eight benchmarks show that A and C scores correlate with performance following a quadratic relationship. By leveraging this relationship, we can identify and train the optimal vision representation for an MLLM, achieving a 99.7% reduction in computational cost without the need for repeated finetuning of the language model. The code is available at https://github.com/bronyayang/Law_of_Vision_Representation_in_MLLMs.

## 1 Introduction

Current multimodal large language models (MLLMs) (Chen et al., 2024a; Liu et al., 2024e;d) have achieved remarkable advancements by integrating pretrained vision encoders with powerful language models (Touvron et al., 2023; Zheng et al., 2023). Among the core components of a general MLLM, vision representation plays a critical role. Many researchers have utilized CLIP (Radford et al., 2021) or SigLIP (Zhai et al., 2023b; Tschannen et al., 2025) as the primary image feature encoder, but their limitations are becoming increasingly noticeable (Tong et al., 2024b; Geng et al., 2023; Yao et al., 2021). As a result, alternative vision representations (Tang et al., 2025) and the combination of multiple vision encoders are being actively explored (Tong et al., 2024a; Lin et al., 2023).

Despite this growing attention, selection of vision representation has largely been empirical. Researchers typically test a set of vision representations on a specific MLLM and choose the one that yields the highest performance on benchmark tasks. This approach, however, is constrained by the number of representations tested and does not address the underlying factors that drive performance differences. As a result, the optimal vision representation for a specific MLLM is often determined by empirical performance rather than a deep understanding of the factors that contribute to success. The question of what fundamentally makes a feature representation achieve the highest performance remains largely unanswered.

To address this gap in understanding what makes a vision representation optimal for MLLMs, we propose the **Law of Vision Representation in MLLMs**. It aims to explain the key factors of vision representation that impact MLLM benchmark performance. Our findings reveal that *cross-modal **Alignment (A)** and **Correspondence (C)** of the vision representation are strongly correlated with model performance.*; specifically, higher A and C lead to improved performance. To quantify this relationship, we define **A and C scores** that measures cross-modal alignment and correspondence in vision representation. The A and C

---

*Corresponding author: xuchenfeng@berkeley.edu

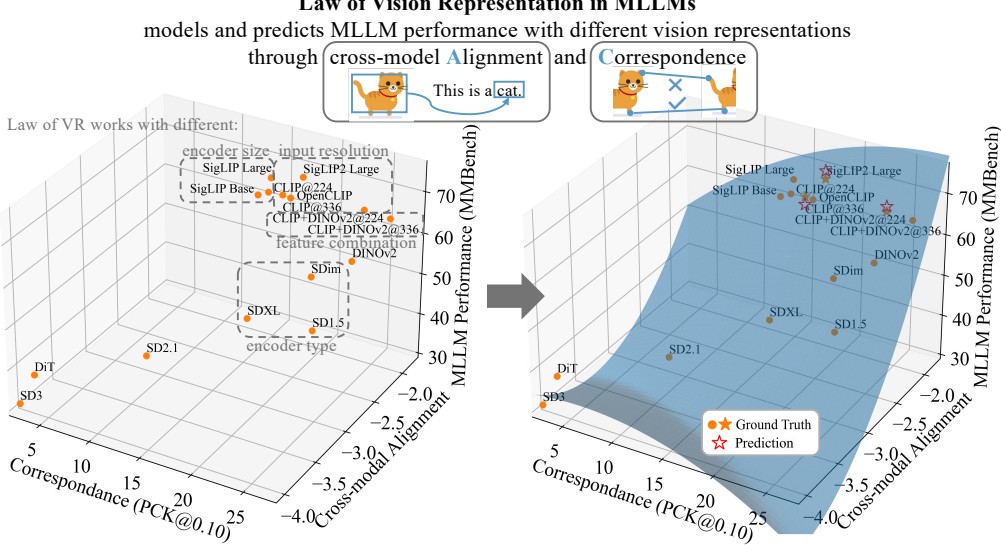

Figure 1: Visualization of the Law of Vision Representation in MLLMs.

scores as well as model performance exhibit a quadratic relationship, with a coefficient of determination of 94.06%.

Furthermore, the Law of Vision Representation guides the selection of an optimal vision representation for MLLMs. Originally, this process was extremely costly because even subtle changes in vision encoding—such as switching encoder types, altering image resolution, or testing feature combinations—require finetuning the language model (Lin et al., 2024). For example, using a top data-efficient MLLM pipeline with a 7B language model requires 3,840 NVIDIA A100 GPU hours to test the 10 encoders, amounting to a cost of approximately \$20,000[1]. Testing additional encoders leads to a linear increase in cost. Moreover, the recent trend of feature combination, which often results in better performance, necessitates combinatorial testing of vision encoders. Testing all possible combinations of 10 encoders results in 1023 combinations, exponentially increasing the cost and energy consumption. This process consumes approximately 100,000 kilowatt-hours[2], enough to drive an electric vehicle around the Earth 13 times.

Thus, we are the first to propose a policy, **AC policy**, that selects the optimal vision representation using AC scores within the desired search space. Unlike traditional methods that rely on benchmarking performance, the AC policy enables the expansion of the search space—allowing for an increased number of vision representations to be considered—without incurring additional costs. We demonstrate that this approach enhances both accuracy and efficiency compared to randomly searching for the optimal representation. The policy successfully identifies the optimal configuration among the top three choices in **96.6%** of cases, with only three language model finetuning across a 15-setting search space.

## 2  Related Works

### 2.1  Vision Representations for MLLMs

Recent studies have explored various vision representations in MLLMs (Beyer et al., 2024; Ge et al., 2024; Liu et al., 2024e; Wang et al., 2024b; Sun et al., 2023; Luo et al., 2024). Interestingly, some findings indicate that relying solely on encoders outside of the CLIP family (Cherti

---

[1]https://replicate.com/pricing

[2]https://www.nvidia.com/content/dam/en-zz/Solutions/Data-Center/a100/pdf/nvidia-a100-datasheet.pdf

et al., 2023; Zhai et al., 2023b; Li et al., 2022), such as DINOv2 (Oquab et al., 2023) and Stable Diffusion (Rombach et al., 2021), often leads to lower performance (Karamcheti et al., 2024; Tong et al., 2024a). However, combining features from these encoders with CLIP features, such as concatenating image embeddings in the channel dimension, significantly enhances performance beyond using CLIP alone (Tong et al., 2024a;b; Liu et al., 2024c; Kar et al., 2024). Researchers intuitively suggest that these additional encoders provide superior detail-oriented capabilities, but no studies have thoroughly analyzed the underlying causes of the performance change (Wei et al., 2023; Lu et al., 2024a). This suggests that the attributes of an optimal vision representation remain not fully understood.

## 2.2 Cross-modal Alignment

Cross-modal alignment refers to the alignment between image and text feature spaces (Duan et al., 2022). This concept emerged with the introduction of text-image contrastive learning (Radford et al., 2021; Jia et al., 2021). Although current MLLMs utilize contrastively pretrained image encoders, the challenge of achieving effective alignment persists (Ye et al., 2024; Zhai et al., 2023a; Woo et al., 2024). Despite efforts to critique the limitations of CLIP family representations and explore alternative vision representations, many approaches continue to rely on contrastively pretrained encoders or adding contrastive loss without fully eliminating them (Zhang et al., 2024b; Lu et al., 2024a; Tong et al., 2024a;b; Liu et al., 2024b). In our work, we point out that alignment in vision representation is essential for improved model performance and is crucial for data efficiency. Without pre-aligned vision representations, extensive data pretraining is required to achieve cross-modal alignment within the language model (Ge et al., 2024; Chen et al., 2024b; Li et al., 2024c).

## 2.3 Visual Correspondence

Visual correspondence is a fundamental component in computer vision, where accurate correspondences can lead to significant performance improvements in tasks, such as image detection (Xu et al., 2024; Nguyen & Meunier, 2019), visual creation (Tang et al., 2023; Zhang et al., 2024c), and MLLMs (Liu et al., 2024a), etc. Correspondences are typically categorized into semantic- and geometric-correspondences. Semantic correspondences (Zhang et al., 2024c; Min et al., 2019) involve matching points that represent the same semantic concept not necessarily representing the same instance. Geometric correspondences (Sarlin et al., 2020; Lindenberger et al., 2023), on the other hand, require matching the exact same point across images, which is often crucial for low-level vision tasks, such as pose estimation (Sarlin et al., 2020; Lindenberger et al., 2023; Zhang & Vela, 2015), and SLAM tasks, etc.

Several studies have pointed out that the CLIP family's vision representation "lacks visual details" (Lu et al., 2024a; Tong et al., 2024b; Ye et al., 2024). We explain this observation through the concept of correspondence. Current MLLMs convert images into embeddings, with each embedding representing a patch of the image. Image features with high correspondence increase the similarity within internal image patches on similar semantics, thereby enabling the retrieval of more detailed information.

## 3 Law of Vision Representation in MLLMs

We introduce the Law of Vision Representation in Multimodal Large Language Models (MLLMs). It states that the performance of a MLLM, denoted as $Z$, can be estimated by two factors: cross-modal alignment ($A$) and correspondence ($C$) of the vision representation, assuming vision representation is the sole independent variable while other components (*e.g.*, language model and alignment module) remain fixed. This relationship can be expressed as:

$$Z \propto f(A, C) \tag{1}$$

where $f$ is a quadratic function of $A$ and $C$.

### 3.1 Assumptions

Following NVLM (Dai et al., 2024), we categorize MLLMs into the following types: (1) Decoder-only MLLMs (Tong et al., 2024a; Liu et al., 2024e; Li et al., 2024a; Liu et al., 2024f; Dai et al., 2024; Lu et al., 2024b; Zhang et al., 2024a; Wang et al., 2024a): These MLLMs consist of vision encoder(s) and an alignment module, such as a multilayer perceptron (MLP), which maps the vision representation into vision tokens. These tokens are designed to have a similar distribution as language tokens and are directly input into a language model in the same manner as language tokens. (2) Cross-attention-based MLLMs (Dai et al., 2024; Bai et al., 2023; Alayrac et al., 2022; Laurençon et al., 2024; Chen et al., 2024c): These MLLMs include vision encoder(s) and an additional module, often serving as a downsampling component, such as a perceiver resampler. The vision tokens generated are integrated into the language model through cross-attention mechanisms.

- The Law of Vision Representation specifically focuses on decoder-only MLLM architecture due to their widespread adoption and their simplicity, which facilitates controlling variables in training recipes and enables clear mathematical modeling.

- We further assume vision representation is the only independent variable, while the alignment module and LLM architecture remain fixed. In the case of a unfrozen vision encoder, we cannot guarantee that the vision encoder does not take the function of the alignment module. This causes the architecture and role of the alignment module to change alongside the encoder, making the experiment uncontrolled and the models no longer comparable.

### 3.2 Theoretical Justification

In this section, we theoretically analyze how an increase in $A$ and $C$ leads to improved model performance. When a vision representation demonstrates high cross-modal alignment and accurate correspondence, the MLLM exhibits the following desired properties:

- *When training a MLLM, if the vision representation is closely pre-aligned with the language distribution, the pretrained language model requires less computational effort to bridge the gap between different modalities during finetuning.* In Section A.1, we provide theoretical justification that *finetuning on well-aligned multimodal data is about equivalent to finetuning on text-only data, eliminating additional effort beyond language finetuning.* This efficiency can lead to improved performance, especially in scenarios where the available training data for finetuning is limited.

- *If the vision representation ensures accurate correspondence, the attention within the image embeddings is precise.* Consequently, the MLLM develops a refined focus on visual content, capturing even details that cannot be derived solely from text-to-image attention, leading to a more detailed interpretation of the image. We provide theoretical justification in Section A.2.

### 3.3 Empirical Justification

In this section, we empirically show that $A$ and $C$ scores are strongly correlated to model performance. To quantify the correlation between $A$ and $C$ as well as model performance, we first propose methods to measure cross-modal alignment and correspondence within the vision representation:

- To quantify cross-modal alignment, we define a metric A SCORE, that measures how well the vision representation is mapped into the language model's space. With both the vision encoder and the LLM frozen, if visual features aligns with the LLM's language space effectively, the LLM's prediction error will be minimized. In other words, a well-aligned vision embedding leads to a higher likelihood for the correct caption tokens.

  Formally, for an input image $I$ and its associated caption $y = (y_1, y_2, \ldots, y_T)$, where $T$ is the sequence length, let $f(I)$ denote the projected visual representation (i.e., the

output of the vision encoder + projector). The conditional probability of generating token $y_t$ is given by:

$$P\big(y_t \mid f(I), y_{<t}\big),$$

, with $y_{<t}$ representing the tokens preceding $y_t$.

The alignment score is then defined as the average log likelihood over all tokens:

$$\text{A SCORE}(I, y) = \frac{1}{T} \sum_{t=1}^{T} \log P\big(y_t \mid f(I), y_{<t}\big)$$

A higher A SCORE indicates that the visual features are more effectively aligned with the language model's representation, as reflected by the increased log-likelihood of the correct caption.

- To quantify correspondence, we measure how accurately key points in one image can be matched to their semantically corresponding locations in another image. Given a pair of image with annotated, semantically matching key points, we first extract features from each image pair. Let $F^s$ and $F^t$ denote the feature maps of the source and target images, respectively.

  Using the feature vectors at the labeled key point positions in $F^s$, we predict the corresponding key points in $F^t$ by selecting the location with the maximum similarity, yielding a set of predicted key points $\{p_1^{pred}, \dots, p_m^{pred}\}$ for $m$ key points. The ground-truth key points for the image pair are denoted by $\{p_1^{GT}, \dots, p_m^{GT}\}$.

  The correspondence score is then defined as the Percentage of Correct Keypoints (PCK), computed as follows:

  $$\text{C SCORE} = \frac{1}{m} \sum_{i=0}^{m} \mathbb{1}_{\left\| p_i^{pred} - p_i^{GT} \right\|_2 < T} \tag{2}$$

  where $T$ is a threshold proportional to the bounding box size of the object in the image, and $\mathbb{1}(\cdot)$ is the indicator function that returns 1 when the condition is satisfied and 0 otherwise.

  A higher C SCORE indicates more accurate key point correspondence, reflecting better vision feature matching.

To capture the overall performance, we integrate the A SCORE and the C SCORE into a single metric called the AC SCORE. We do this by fitting a second-degree polynomial to benchmark performance, which allows us to model potential nonlinear interactions between $A$ and $C$. Formally, the AC Score is defined as:

$$\text{AC SCORE} = \sum_{\alpha=0}^{2} \sum_{\beta=0}^{2-\alpha} w_{\alpha\beta} A^\alpha C^\beta \tag{3}$$

where $w_{\alpha\beta}$ are trainable parameters that are optimized to best fit the benchmark performance.

This formulation allows the AC Score to capture both the individual contributions of alignment and correspondence, as well as their interaction effects.

**Results.** We fit a simple regression model using 15 vision representations across 4 vision-based MLLM benchmarks. As shown in Figure 2, the average coefficient of determination ($R^2$) obtained is 94.06% when using the AC score of the vision representations. For comparison, we also fit models using 15 random scores, the A score alone, and the C score alone, all with quadratic functions. The random scores and single-factor models show lower correlations with performance. This result highlights *the strong correlation between the AC score and MLLM performance, validating the Law of Vision Representation*. Refer to Section 5.4 for details.

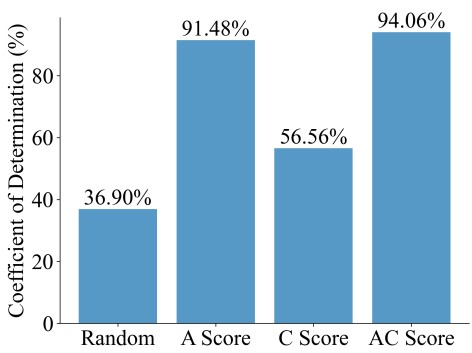

Figure 2: $R^2$ values for regression models fitted on various scores.

## 4 AC Policy

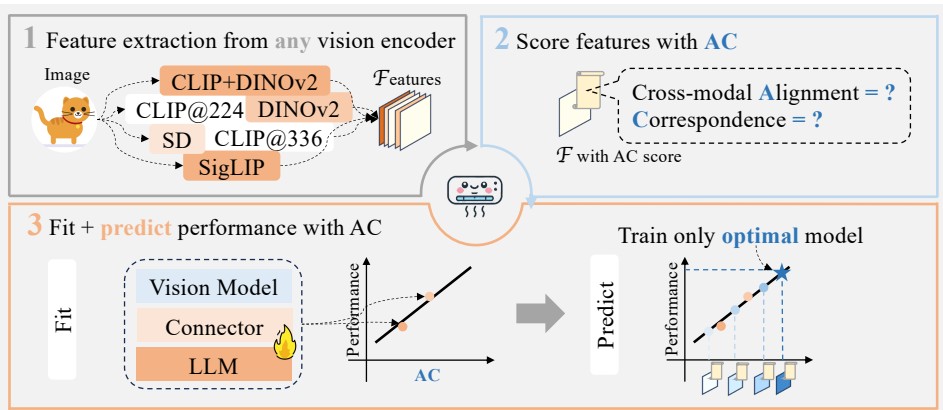

Figure 3: Overall framework of AC policy.

**Problem Formulation.** The MLLM architecture assumed in this framework consists of a frozen vision encoder, followed by a trainable connector (alignment module) and the pretrained language model. To determine the optimal out of $k$ vision representations for the MLLM, we originally needs finetune LLM $k$ times, making the scaling of $k$ difficult. Therefore, we propose AC policy, as illustrated in Figure 3, to efficiently estimate the optimal vision representation from a search space consisting of $k$ vision representations. We finetune only $k'$ LLMs to obtain downstream performance, allowing $k$ to scale without significant cost, where $k' \ll k$. The value of $k'$ should be determined based on the computational budget allocated for vision representation selection.

**Policy Fitting.** Let $\mathbf{X} \in \mathbb{R}^{k \times 6}$ be the matrix containing AC scores of vision representation in the search space. We subsample $k'$ data points from $\mathbf{X}$, denoted as $\mathbf{X}_s \in \mathbb{R}^{k' \times 6}$, to serve as the input to the regression model:

$$\mathbf{y} = \mathbf{X}_s \mathbf{w} + \epsilon \tag{4}$$

Here, $\mathbf{w} \in \mathbb{R}^6$ is the vector of model parameters, $\epsilon \in \mathbb{R}^{k'}$ is the vector of error terms, and $\mathbf{y} \in \mathbb{R}^{k'}$ represents the downstream performance on a desired benchmark.

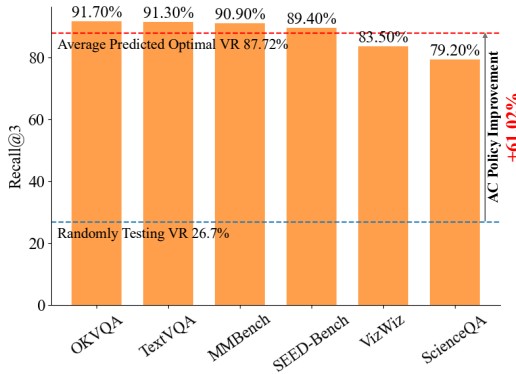

Figure 4: Given a limited budget of 4 finetunings, AC policy achieves 87.72% Recall@3 in predicting the optimal vision representation.

| Vision Representation | Resolution |
|---|---|
| *Single vision encoder: feed-forward models* | |
| OpenAI CLIP ViT-L/14 | 224 |
| OpenAI CLIP ViT-L/14 (Radford et al., 2021) | 336 |
| OpenCLIP ViT-L/14 (Cherti et al., 2023) | 224 |
| DINOv2 ViT-L/14 (Oquab et al., 2023) | 224 |
| SigLIP ViT-B/16 (Zhai et al., 2023b) | 224 |
| SigLIP ViT-L/16 (Zhai et al., 2023b) | 256 |
| SigLIP2 ViT-L/16 (Tschannen et al., 2025) | 256 |
| *Single vision encoder: diffusion models* | |
| SD 1.5 (Rombach et al., 2022) | 768 |
| SD 2.1 (Rombach et al., 2022) | 768 |
| SD Image Variations | 768 |
| SD XL (Podell et al., 2023) | 512 |
| DiT (Peebles & Xie, 2023) | 512 |
| SD 3 (Esser et al., 2024) | 512 |
| *Multiple vision encoders: feature combination* | |
| CLIP+DINOv2 ViT-L/14 | 224 |
| CLIP+DINOv2 ViT-L/14 | 336 |

Table 1: Vision representations explored.

**Sampling Strategy.** The selection of $k'$ can impacts the function fit and, consequently, the accuracy of predictions. To avoid sampling points that are too close in terms of their A and C scores, we employ a sampling strategy based on the coordinates.

The normalized A and C score pairs of $k$ vision representation can be plotted on a 2D graph as coordinates $(A, C)$, To ensure diverse sampling, we divide the graph into regions. For each iteration $j$ in which the total sampled points do not yet fulfill $k'$, we divide the graph into $4^j$ equal regions. We then remove empty regions and those that contain previously sampled points. The next data point is randomly selected from a remaining region.

**Results.** In Figure 4, we demonstrate that the *AC policy consistently predicts the optimal vision representation using minimal resources* within a finite search space of 15 settings. Our aim is to finetune only a small subset of this space while ensuring that the optimal vision representation is among the top-3 predictions (Recall@3). With a computational budget equivalent to 4 full finetuning runs, a random subset selection achieves only 26.7% Recall@3. In contrast, the AC policy achieves 87.72% Recall@3 (averaged over 6 benchmarks) while still requiring just 4 full training runs. For further details, see Section 5.5.

# 5 Empirical Result Details

## 5.1 Experiment Settings

For our MLLM pipeline, we deploy LLaMA-based LLM, specifically Vicuna-7B 1.5 (Zheng et al., 2023) as well as Qwen-14B 2.5 Instruct (Qwen et al., 2025), and utilize a widely used 2-layer GeLU-MLP connector as the projector. For vision representation, we explore a variety of encoder types and sizes, input resolutions, training paradigms, and feature combinations, as detailed in Table 1.

Our training process consists of two stages. In Stage 1, we perform alignment using the LLaVA 1.5 dataset with 558K samples (Liu et al., 2024e), training only the projector. In Stage 2, we train both the connector and the language model on an expanded LLaVA 1.5 dataset containing 665K samples.

The MLLM benchmarks used in this paper include four vision-based benchmarks (MM-Bench (Liu et al., 2023), MME (Fu et al., 2023), OKVQA (Marino et al., 2019), SEED-Bench (Li et al., 2024b)) and four QCR-based benchmarks (MMMU (Yue et al., 2024), TextVQA (Singh et al., 2019), VizWiz (Gurari et al., 2018), ScienceQA (Lu et al., 2022)).

## 5.2 AC Score

To compute the cross-modal alignment score, we perform Stage 1 training on all vision representations to obtain the projected vision representations. This stage requires significantly less computation than Stage 2, involving only 0.298% of the trainable parameters. The image–caption pairs are taken from the LLaVA-558K dataset, and the alignment score is computed by averaging the results across 100 randomly sampled images.

For the correspondence score, we follow common practices using the SPair-71k dataset (Min et al., 2019). Note that all benchmarks share the same A and C scores, while the quadratic parameters in the function fitting adapt to capture variations across tasks.

## 5.3 Feature Extraction

Both MLLM training and score computation involve image feature extraction. Below, we introduce the approach for obtaining two types of vision representations.

**Vision Representation from Feed-forward Models.** Given an image $I \in \mathbb{R}^{H \times W \times 3}$ we process it either in its raw form for U-Net models or in a patchified form for transformer models. For transformers, we extract the last hidden state $F \in \mathbb{R}^{l \times c}$ where $l$ is the sequence length and $c$ is the hidden dimension. In the case of the U-Net model, we take the intermediate activation $F \in \mathbb{R}^{\hat{H} \times \hat{W} \times c}$ after the first upsampling block. Note that the features from these two types of models are interchangeable between sequence and grid formats through reshaping and flattening. For consistency, the following sections assume that all features have been pre-converted into the same format.

**Vision Representation from Diffusion Models.** Diffusion model is primarily used for generating images via multi-step denoising, yet a recent trend is to use diffusion model as the vision representation model (Xu et al., 2024; 2023; Zhang et al., 2024c; Tong et al., 2024a). Specifically, for diffusion models, given an image $I \in \mathbb{R}^{H \times W \times 3}$, we first add noise to the VAE-encoded representation of $I$:

$$x_t = \sqrt{a_t} \cdot \text{VAE}(I) + (\sqrt{1 - a_t}) \cdot \epsilon \tag{5}$$

where $\epsilon \sim \mathcal{N}(0, \mathbf{I})$ and $a_t$ is determined by the noise schedule. Note that we utilize the little-noise strategy by setting the $t = 1$. In that case, the diffusion model only denoises the noise-latents once and we treat the one-step denoising latents as the vision representation features.

## 5.4 Additional Results on the Law of Vision Representation

In Section 3, we demonstrate the strong correlation between the AC score and MLLM performance by analyzing the coefficient of determination ($R^2$) obtained from fitting a quadratic regression model. In this section, we further ablate the experiments by adding baselines, fitting model performance with random scores, A scores, and C scores separately. Additionally, we explored the relationship between the A score and C score by fitting a linear regression model. We provide more insights about the AC relationship in appendix A.3. Besides, we avoid higher-degree transformations to prevent overfitting, which could obscure the true relationship between A and C scores.

As shown in Table 2, the results indicate that using the AC score consistently outperforms all other set-

| Fitting Data | $R^2$ (Vision) | $R^2$ (OCR) |
|---|---|---|
| *No transformation on fitting data* | | |
| Random | 4.03% | 1.75% |
| A Score | 80.53% | 58.00% |
| C Score | 39.02% | 14.57% |
| AC Score | 80.55% | 62.06% |
| *Polynomial transformation on fitting data* | | |
| Random | 36.90% | 31.26% |
| A Score | 91.48% | 79.67% |
| C Score | 56.56% | 30.11% |
| AC Score | 94.06% | 83.85% |

Table 2: Averaged $R^2$ results of AC and other baselines fitting on MLLM benchmarks.

tings in terms of $R^2$ values. While this observation holds regardless of the degree of fitted function, using a second-degree polynomial on A and C scores yields the highest correlation with model performance. This suggests an inherent trade-off between A and C scores: vision representations with high cross-modal alignment often exhibit lower correspondence, and vice versa.

Interestingly, we observe a lower correlation between OCR-based benchmark performance and C scores, which leads to a reduced correlation between the AC score and OCR-based benchmark performance. In Section 6, we discuss how the use of the SPair-71k correspondence dataset across all benchmarks fails to adequately capture correspondence in images containing text.

Here, we additionally provide MLLM with Qwen 2.5 14B Instruct as LLM backbone fitting results, as shown in Table 3. Specifically, we use the exact same settings (data, vision representations, hyperparameters), changing only the LLM backbone to Qwen 2.5 14B. We again evaluate the resulting MLLMs on 8 benchmarks, recompute the A and C scores, reporting the fitting $R^2$. The averaged fitting $R^2$ are above 90% for vision based benchmarks and above 80% for OCR based benchmarks, comparable to Vicuna 1.5 7B, showing that our conclusion applies to different types and sizes of LLM backbones.

| LLM Backbone | $R^2$ **(Vision)** | $R^2$ **(OCR)** |
|---|---|---|
| Vicuna 1.5 7B | 94.06% | 83.85% |
| Qwen 2.5 14B | 91.50% | 82.27% |

Table 3: Averaged law fitting $R^2$ comparison for different types and sizes of LLM backbones.

## 5.5 Additional Results on the AC Policy

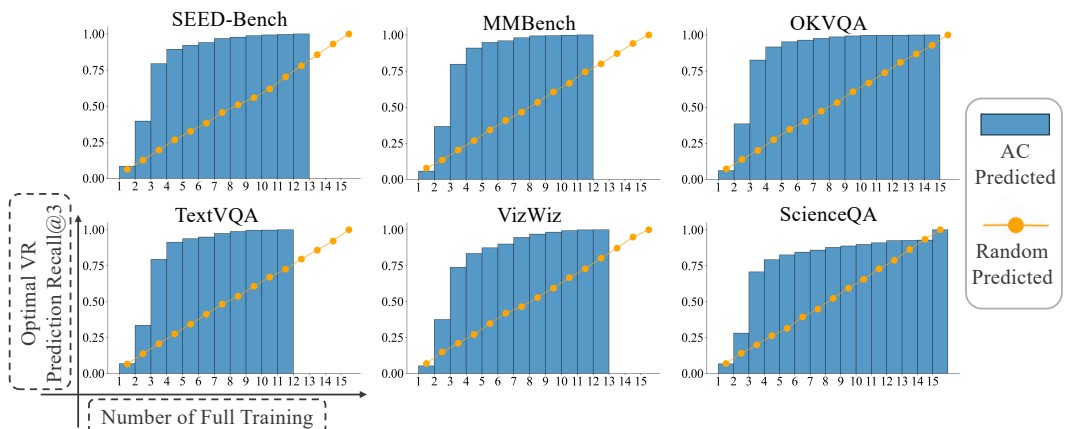

Figure 5: Number of full training (LLM finetuning) cycles required to include the optimal vision representation within the top-3 predictions (Recall@3).

In Section 4, we demonstrate that fitting the AC score consistently predicts the optimal vision representation with minimal resources, given a finite search space—in this case, 15 settings. In this section, we provide detailed visualization for Figure 4.

When performing ablation experiments on vision encoders, it's common to randomly select a subset to train on. However, as shown in Figure 5, with 1000 runs of simulated ablation experiments, we found that to include the optimal vision representation 85.6% of the time, at least 13 out of the 15 settings need to be trained. This suggests that running a small subset of vision representations is unreliable, especially as the search space expands, making it increasingly unlikely to identify the true optimal representation by training only a subset.

In contrast, the AC policy requires only 4 full training runs on average to reach 87.72% Recall@3. For the most successful prediction benchmark, OKVQA, the policy successfully identifies the optimal configuration among the top three choices in 91.7% of cases, with only four language model finetuning runs across a 15-setting search space. This result shows that

AC policy significantly reduces the effort and cost of exploring vision representations for MLLMs.

Obtaining the AC policy results in significant computational cost reduction. After fitting the AC function, testing a new vision encoder only requires MLLM Stage 1 training (i.e. training a small projector), while traditional full MLLM post-training requires training both the projector and LLM. Concretely, we quantify computational cost using the number of trainable parameters, which is commonly correlated with both memory and compute. A decoder-only MLLM's Stage 1 trains a small two-layer MLP, while Stage 2 involves fine-tuning a large language model with 7B parameters. This leads to a parameter ratio of about 0.003, yielding a 99.7% reduction.

# 6 Limitation

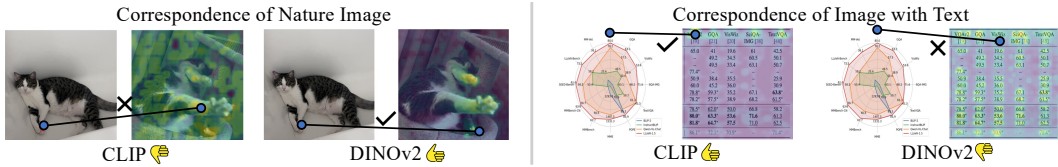

Figure 6: Visualization of correspondence on natural images and images containing text for CLIP and DINOv2.

We find that OCR-based benchmarks correlate less with the AC score than vision-based ones, making MME and MMMU outliers. For example, SigLIP2-Large@256 outperforms CLIP-Large@336 in both alignment (-1.81 vs. -1.97) and correspondence (16.75 vs. 15.66) but underperforms on MME due to OCR-heavy categories.

This discrepancy arises because vision representations exhibit different correspondence accuracy on different domain of images, as shown in Figure 6. The SPair-71k dataset (for the C score) focuses on natural images (e.g., cats, trains), whereas CLIP excels at text-based tasks not captured in SPair-71k. Likewise, our alignment score, calculated using LLaVA 558K (a subset of LAION, CC, SBU), lacks sufficient OCR, numeric, and symbolic data. Consequently, both the A and C scores underrepresent performance on OCR-related tasks.

To our knowledge, an OCR-specific correspondence dataset does not currently exist, and systematically designed OCR short caption datasets are scarce. We intend to pursue further investigation in this direction and encourage other researchers to do the same, as advancements in this area would be valuable for the broader MLLM community—particularly for tasks requiring the understanding of tables and charts, a fundamental capability.

# 7 Conclusion

In this work, we present the Law of Vision Representation in MLLMs, uncovering a strong and quantifiable relationship between cross-modal alignment, feature correspondence, and overall model performance. Our proposed A and C scores capture this relationship with high fidelity, enabling a principled way to reason about vision representation quality. Through extensive evaluation, we demonstrate that model performance exhibits a clear quadratic correlation with AC scores, providing not only an interpretable diagnostic but also a predictive signal for selecting optimal visual features.

By leveraging this insight, we introduce the AC policy, a cost-efficient strategy to identify high-performing vision representations without repeatedly finetuning the language model. This yields up to 99.7% reduction in compute cost, transforming what was previously an expensive and combinatorially large search problem into a tractable one. Our findings offer a new perspective on vision-language alignment and open the door to more scalable MLLM development.

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

# A    Appendix

## A.1    Theoretical Justification of Vision Representation with High Cross-modal Alignment

In Section 3.2, we state that when training an MLLM, if the vision representation is closely pre-aligned with the language distribution, then the pretrained language model requires less computational effort to bridge the gap between different modalities during finetuning. In this section, we show that using well-aligned vision representation, finetuning on multi-modal data is about equivalent to finetuning on text-only data, eliminating additional effort beyond language finetuning.

Assume the vision embedding distribution $D_{image}$ and text embedding distribution $D_{text}$ are well-aligned in the MLLM. For a shared concept $c$, the image embedding after the alignment module and its corresponding text embedding, $E_c^{image} \sim D_{image}$ and $E_c^{text} \sim D_{text}$, are close in distance, meaning:

$$\|E_c^{image} - E_c^{text}\| \leq \epsilon \tag{6}$$

where $\epsilon$ is a small constant. Given this condition, we can show that the output of the MLLM with multimodal embeddings $[E_c^{image}, E_1, E_2, \ldots, E_n]$ is close to the output with text-only embeddings $[E_c^{text}, E_1, E_2, \ldots, E_n]$.

Since our language model $f$ is well-trained and pre-normed, the input space to each transformer layer is bounded and compact, meaning that the values of the input are bounded by a small constant. This implies that the continuously differentiable function $f$ is Lipschitz (Kim et al., 2021). This property ensures that small changes in the input of the language model of the MLLM result in small, controlled changes in the output:

$$
\begin{aligned}
\Big\| f\big([E_c^{image}, E_1, E_2, \ldots, E_n]\big) &- f\big([E_c^{text}, E_1, E_2, \ldots, E_n]\big) \Big\| \\
&\leq L \Big\| [E_c^{image}, E_1, E_2, \ldots, E_n] - [E_c^{text}, E_1, E_2, \ldots, E_n] \Big\| \\
&\leq L\epsilon.
\end{aligned}
\tag{7}
$$

where $L$ is the Lipschitz constant. This closeness in output distance implies that even with multimodal data, the pretrained language model mimics the training dynamics closely resemble language-only finetuning.

## A.2    Theoretical Justification of Vision Representation with Accurate Correspondence

In Section 3.2, we state that if the vision representation ensures accurate correspondence, the attention within the image embedding is precise. In this section, we show that vision representation with accurate correspondence can help vision information retrieval in the attention mechanism. Therefore, more visual details are considered even if not attended by the text token.

Consider an input $[E_0^{image}, E_1^{image}, E_2, \ldots, E_n]$ to the transformer, where the image embeddings $E_0^{image}$ and $E_1^{image}$ are derived from different patch of a high correspondence vision representation. By definition, the dot product $E_0^{image} \cdot E_1^{image}$ is large if the two corresponding original image patches share related information.

Suppose a text token $E_2$ attends to $E_0^{image}$. We show that it is also able to retrieve $E_1^{image}$ and vice versa. This can be demonstrated as follows:

$$\text{score}(E_2, E_0^{image}) = \frac{(E_2 W^Q) \cdot (E_0^{image} W^K)}{\sqrt{d_k}} \tag{8}$$

If $\text{score}(E_2, E_0^{\text{image}})$ is high, and $(E_0^{\text{image}} W^K)^\top (E_1^{\text{image}} W^K)$ is also large (assuming $W^K$ does not distort the vectors drastically), then by transitivity, $\text{score}(E_2, E_1^{\text{image}})$ is also likely to be high. This transitivity ensures that attention is effectively spread across related visual information, enhancing the model's ability to interpret visual content in greater detail.

### A.3 Analysis of Quadratic Relationship between A and C

We first provide an intuitive explanation of why we choose to capture a quadratic relationship between cross-modal alignment and correspondence. Then, we analysis the coeffcient of the fitted function to reveal a more in-depth relationship between these two factors.

Here's our insight:

- Good semantic correspondence means the vision part of MLLM can really tell what things are in a picture, even if they look a bit different. Like knowing all dogs are dogs, no matter their breed. But this alone isn't enough for the MLLM to "talk" about them. Those visual details need to be understood by the LLM.
- Good language alignment means the visual information is perfectly understood by the language model. But if the initial visual information isn't very good at telling different things apart (poor semantic correspondence), then aligning it perfectly won't help much when the MLLM sees something new or slightly different.

So, for MLLMs to work best, both semantic correspondence and language alignment need to work together. More importantly, they interplay with each other more than just adding up:

If the visual features are already well-aligned with language (like in models such as CLIP), it actually helps the MLLM understand the semantic details better. That's because if you can map "dog" to the word "dog" consistently, it reinforces the concept of "dog" in the visual world.

But why is this interplay represented in a quadratic form (like a hill)? Think of it in this way:

If language alignment is very weak, even if you make semantic correspondence a little better, the MLLM's performance won't jump much. It's like trying to have a good conversation when you don't speak the same language very well. Take a model like DINOv2. It's great at seeing things, but it might not be as good at linking those visual details to language as, say, CLIP. In this case, making the visual part even better won't help MLLM performance much because the language alignment is the bottleneck.

As language alignment improves, the MLLM can then really benefit from better semantic correspondence. It's like they're working together to reach that "sweet spot" where the performance really shines. Both getting better at the same time, or one unlocking the potential of the other, leads to this curved, quadratic rise in performance.

In short, quadratic models include $A^2$ and $C^2$ and interaction term $\alpha_{AC}$ which includes exactly the relationship we want to capture. In our paper, we test linear, second-degree polynomial relationships. Our result shows that the second-degree relationship fitted the best among others while still maintaining simplicity. More complex functional forms would obscure the clear individual contribution of A and C, as well as interaction between alignment and correspondence.

Next, We discuss the interation alignment and correspondence from two aspects: the ideal relationship between A and C suggested by the law, and the actual relationship we observe from existing encoders. To provide the ideal relationship, we extract the cross-term coefficient ($\alpha_{AC}$) of the fitted functions as shown in Table 4.

We observe two kind of interactions:

**Synergy** ($\alpha_{AC} \gg 0$): MMBench, MME, TextVQA, VizWiz, ScienceQA, and SeedBench all have $\alpha_{AC} > 0$. For these six tasks, performance improves more than additively when both $A$ and $C$ increase.

**Mild trade-off** ($\alpha_{AC} < 0$): MMMU and OKVQA show $\alpha_{AC} < 0$, indicating a slight penalty if $A$ and $C$ are both pushed upward (i.e., diminishing returns).

These results demonstrate that on most benchmarks, alignment and correspondence actually reinforce each other (positive interaction). Therefore, $A$ and $C$ strengthen each other, so the ideal scenario would be increasing both to achieve the best performance.

| Benchmark | $\alpha_{AC}$ |
|---|---|
| MMBench | +1.1301 |
| MME | +1.5819 |
| MMMU | -0.4872 |
| OKVQA | -0.0364 |
| TextVQA | +4.9804 |
| VizWiz | +2.9964 |
| ScienceQA | +4.4644 |
| SeedBench | +0.5772 |

Table 4: Interaction term $\alpha_{AC}$ of fitted function for all eight benchmarks.

However, from our observations of $A$ and $C$ of vision encoders, the factors often have a trade-off: Intuitively, $A$ measures how well image features align to a shared text embedding, whereas $C$ captures intra-image consistency (e.g., visual cluster compactness). Forcing very high alignment often collapses visual cluster structure (hurting correspondence). In practice, looking at the $A$ and $C$ scores in Appendix Section A.5, common visual encoders often have high $A$ but low $C$ (CLIP and SigLIP) or low $A$ but high $C$ (DINOv2 and Diffusion features).

### A.4 All Settings Benchmark Performance

In this section, we present the performance results of all 15 vision representation settings, as summarized in Table 5. The benchmarks we evaluated include:

- MMBench (Liu et al., 2023): A set of multiple-choice questions designed to assess 20 different ability dimensions related to perception and reasoning.

- MME (Fu et al., 2023): A dataset focused on yes/no questions, covering areas such as existence, counting, position, and color, primarily based on natural images.

- MMMU (Yue et al., 2024): Multiple-choice questions targeting college-level subject knowledge and deliberate reasoning, primarily testing the language model's abilities.

- OKVQA (Marino et al., 2019): Open-ended questions based on the MSCOCO (Lin et al., 2014) dataset, spanning 10 different knowledge categories.

- TextVQA (Singh et al., 2019): Open-ended questions designed to evaluate the model's OCR capabilities.

- VizWiz (Gurari et al., 2018): Open-ended questions sourced from people who are blind, aimed at testing the model's OCR capabilities.

- ScienceQA (Lu et al., 2022): A multiple-choice science question dataset, with 86% of the images being non-natural, covering topics in natural science, social science, and language science.

- SEED-Bench (Li et al., 2024b): A benchmark consisting of multiple-choice questions designed to assess both spatial and temporal understanding.

### A.5 All Settings AC Scores

We provide the AC scores of all 15 vision representation settings, as summarized in Table 6.

### A.6 More Visualization of Correspondence

We provide additional visualizations of correspondence for four different vision representations: CLIP, SigLIP, DINOv2, and Stable Diffusion 1.5. Figures 7 and 8 display pairs

|  | CLIP@336 | CLIP@224 | OpenCLIP | SigLIP Base | SigLIP Large |
|---|---|---|---|---|---|
| MMBench | 64.26 | 64.18 | 63.40 | 61.86 | 65.46 |
| MME | 1502.70 | 1449.64 | 1460.28 | 1425.00 | 1455.18 |
| MMMU | 35.0 | 36.2 | 37.2 | 35.8 | 36.6 |
| OKVQA | 53.20 | 56.13 | 56.36 | 54.01 | 57.02 |
| TextVQA | 46.04 | 42.67 | 40.13 | 36.00 | 42.44 |
| VizWiz | 54.27 | 51.69 | 52.11 | 53.17 | 51.49 |
| ScienceQA | 69.26 | 68.82 | 67.87 | 66.88 | 70.20 |
| SEED-Bench | 66.09 | 65.13 | 64.71 | 64.40 | 66.28 |
|  | C+D@224 | C+D@336 | SigLIP2 Large | DINOv2 | DiT |
| MMBench | 65.72 | 65.12 | 67.35 | 58.50 | 33.68 |
| MME | 1436.42 | 1475.19 | 1486.66 | 1295.47 | 902.00 |
| MMMU | 36.9 | 34.6 | 34.2 | 34.6 | 32.7 |
| OKVQA | 55.94 | 56.92 | 57.57 | 54.78 | 33.75 |
| TextVQA | 40.04 | 46.17 | 47.2 | 14.27 | 10.82 |
| VizWiz | 54.04 | 53.44 | 56.55 | 49.67 | 49.92 |
| ScienceQA | 69.11 | 67.63 | 69.41 | 65.15 | 63.46 |
| SEED-Bench | 65.39 | 66.38 | 68.22 | 61.39 | 40.66 |
|  | SDXL | SD3 | SD2.1 | SD1.5 | SDim |
| MMBench | 43.73 | 32.82 | 28.87 | 42.53 | 52.84 |
| MME | 1212.69 | 843.43 | 905.27 | 1163.90 | 1205.33 |
| MMMU | 32.8 | 32.4 | 32.8 | 33.9 | 33.7 |
| OKVQA | 41.78 | 34.95 | 34.41 | 39.14 | 46.04 |
| TextVQA | 11.81 | 10.77 | 10.46 | 11.64 | 13.77 |
| VizWiz | 47.14 | 47.12 | 46.59 | 50.14 | 47.33 |
| ScienceQA | 65.25 | 62.27 | 62.67 | 63.31 | 66.34 |
| SEED-Bench | 53.78 | 38.94 | 38.82 | 50.00 | 50.33 |

Table 5: Benchmark performance of all 15 settings. C+D means feature combination of CLIP and DINOv2. The table provides data points for function fitting and is not intended for comparison.

|  | CLIP@336 | CLIP@224 | OpenCLIP | SigLIP-B | SigLIP-L |
|---|---|---|---|---|---|
| *Correspondence* |  |  |  |  |  |
| PCK@0.10 | 15.66 | 14.30 | 16.22 | 12.89 | 13.66 |
| *Cross-modal Alignment* |  |  |  |  |  |
| Log likelihood | -1.97 | -1.98 | -1.93 | -1.92 | -1.83 |
|  | C+D@224 | C+D@336 | SigLIP2-L | DINOv2 | DiT |
| *Correspondence* |  |  |  |  |  |
| PCK@0.10 | 23.62 | 26.08 | 16.75 | 24.51 | 1.91 |
| *Cross-modal Alignment* |  |  |  |  |  |
| Log likelihood | -1.96 | -1.95 | -1.81 | -2.32 | -3.76 |
|  | SDXL | SD3 | SD2.1 | SD1.5 | SDim |
| *Correspondence* |  |  |  |  |  |
| PCK@0.10 | 16.52 | 3.09 | 6.99 | 22.02 | 20.90 |
| *Cross-modal Alignment* |  |  |  |  |  |
| Log likelihood | -2.69 | -4.13 | -2.81 | -2.53 | -2.37 |

Table 6: AC scores of all 15 settings. C+D means feature combination of CLIP and DINOv2. The table provides data points for function fitting and is not intended for comparison.

of source-target images for each of the four vision representations. In each pair, the left image is the source, and the right image is the target. The red dot on both images indicates the predicted key points using the vision representation. Ideally, these key points should correspond to the same semantic meaning. For example, a red dot on the "left cat ear" in the source image should correspond to the "left cat ear" in the target image. The green areas highlight regions of relatively high similarity with the source points.

In Figure 7, DINOv2 demonstrates superior correspondence for natural images compared to the other vision representations. It accurately matches small parts of the cat between the left and right images, whereas CLIP struggles to correctly identify and align features such as left, right, front, and back.

In Figure 8, , the CLIP family shows precise correspondence for text within images. For instance, when the source image points text like "LLaVA" or "VQAv2", CLIP accurately matches all instances of the text in the target image. In contrast, other vision representations known for "accurate correspondence" in computer vision, such as DINOv2 and Stable Diffusion, fail to provide the same level of accuracy when dealing with images containing text. This emphasizes a key distinction in selecting vision representations for computer vision tasks versus multimodal large language models (MLLMs).

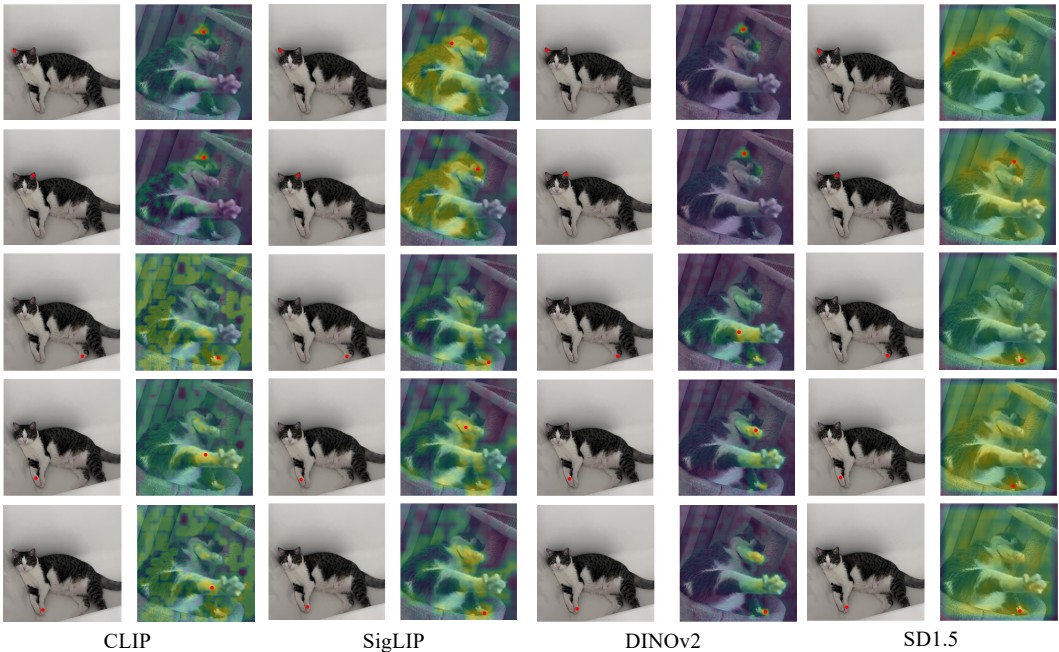

| CLIP | SigLIP | DINOv2 | SD1.5 |

Figure 7: Correspondence of natural images for different vision representations.

### A.7 Pseudo Code

Computing the A score is a simple loss calculation using the frozen MLLM; therefore, we provide pseudocode for the other algorithms, including the computation of the C score at Algorithm 1, region-based sampling at Algorithm 2, and the AC policy at Algorithm 3.

### A.8 Limitation of AC Policy

Figure 9 shows two benchmarks—MME and MMMU—where the AC policy fails to predict the optimal vision representation. For details on the reason of this behavior, please refer to Section 6.

### A.9 AC Policy Recall@1

---

**Algorithm 1:** COMPUTE C SCORE

---

**Input:** Set of paired images with key points $S$ from SPair-71k;
Vision encoder $E$;
Threshold $T$.
**Output:** $C$ score for vision encoder $E$.

```
// Initialize correspondence lists
```
$G \leftarrow []$ ;                                    `// Ground truth keypoint correspondences`
$P \leftarrow []$ ;                                    `// Predicted keypoint correspondences`

**foreach** $(I_1, K_1, I_2, K_2) \in S$ **do**
  `// Extract feature representations`
  $F_1 \leftarrow E(I_1)$;
  $F_2 \leftarrow E(I_2)$;

  `// Compute similarity matrix`
  $S_{\text{sim}} \leftarrow F_1 \cdot F_2^T$;

  `// Transform keypoints from` $I_1$ `to` $I_2$
  $\hat{K}_2 \leftarrow \text{calculate\_keypoint\_transformation}(S_{\text{sim}}, K_1)$;

  `// Store ground truth and predicted keypoints`
  $G.\text{append}(K_2)$;
  $P.\text{append}(\hat{K}_2)$;

```
// Compute correctness score
```
$E_{\text{error}} \leftarrow \text{Euclidean\_distance}(P, G)$;
$C_{\text{correct}} \leftarrow \text{sum}(E_{\text{error}} < T)$;
$C_{\text{score}} \leftarrow \frac{C_{\text{correct}}}{\text{total keypoints in } K_2}$;

---

**Algorithm 2:** REGION-BASED SAMPLING

---

**Input:** $k$ A and C score pairs from models $ACs$; *past_sampled* models; current sampling
*level* (1 to $k'$, increments when regions are exhausted as each region is sampled
only once)
**Output:** Sampled *model* to train next
*regions* $\leftarrow \{\}$;
**for** $AC \in ACs$ **do**
  *region_key* $\leftarrow \text{determine\_region}(A, C, level)$ ; `// Identify the region based on A`
    `and C coordinates`
  *regions*[*region_key*].*append*((*model*, $A, C$));
Remove models in *past_sampled* from *regions*;
*remaining_regions* $\leftarrow$ keys of *regions*;
*chosen_region* $\leftarrow$ randomly select from *remaining_regions*;
*model* $\leftarrow$ randomly select from *regions*[*chosen_region*];

---

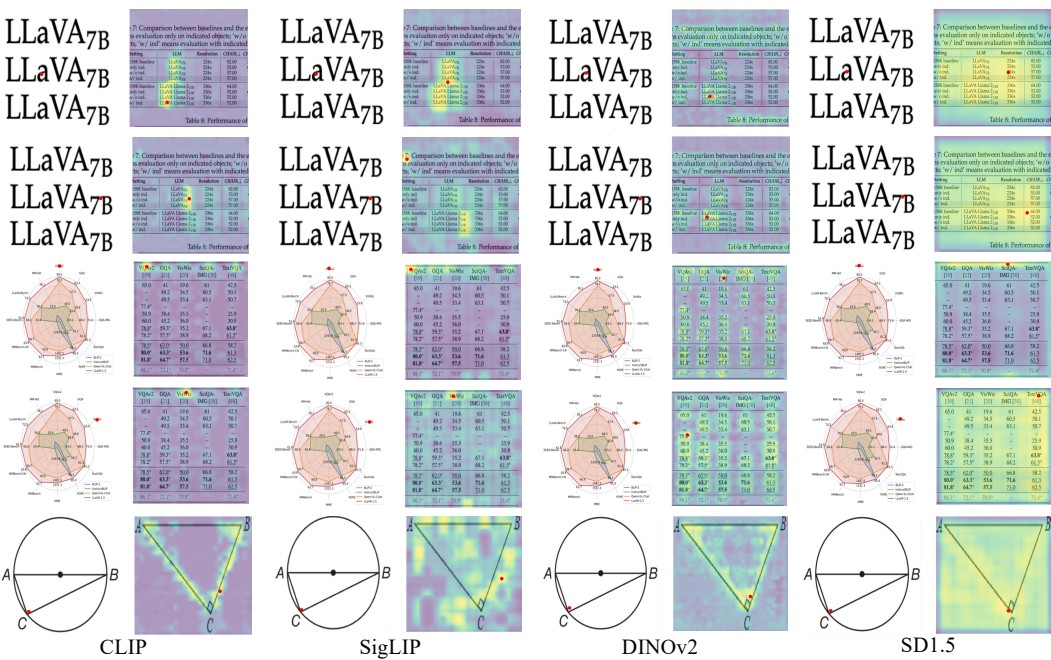

Figure 8: Correspondence of images with text for different vision representations.

---

**Algorithm 3: AC POLICY**

**Input:** $k$ vision encoders with pretrained projectors $V$; computation budget $k'$
**Output:** A ranking of $k$ MLLMs based on performance
$ACs \leftarrow [(\texttt{Compute\_A\_Score}(v), \texttt{Compute\_C\_Score}(v)) \mid v \in V]$;
$past\_sampled \leftarrow []$;
$train\_ACs \leftarrow []$;
$train\_performance \leftarrow []$;
**for** $i \leftarrow 1$ **to** $k'$ **do**
 $model \leftarrow \texttt{Region\_based\_Sampling}(ACs, past\_sampled)$;
 $performance \leftarrow$ Fully train $model$;
 $train\_ACs.\text{append}(\text{AC of } model)$;
 $train\_performance.\text{append}(performance)$;
 $past\_sampled.\text{append}(model)$;

$poly \leftarrow \text{PolynomialFeatures}(degree = 2)$;
$transformed\_train\_ACs \leftarrow poly.\text{fit\_transform}(train\_ACs)$;
$regression \leftarrow \text{LinearRegression}()$;
$regression.\text{fit}(transformed\_train\_ACs, train\_performance)$;
$ranking \leftarrow \text{Rank } V \text{ by regression predictions on } ACs$;

---

In Section 5.5, we used Recall@3 as a metric for AC policy effectiveness to capture potential performance fluctuations during MLLM training and evaluation. Our AC policy's core aim is to narrow the search space to a small set of promising vision encoders, rather than pinpointing a single best encoder. Recall@3 directly quantifies whether the true top-performers appear within our top-3 recommendations, which:

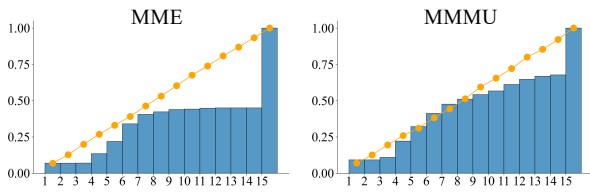

Figure 9: Limitation of AC policy.

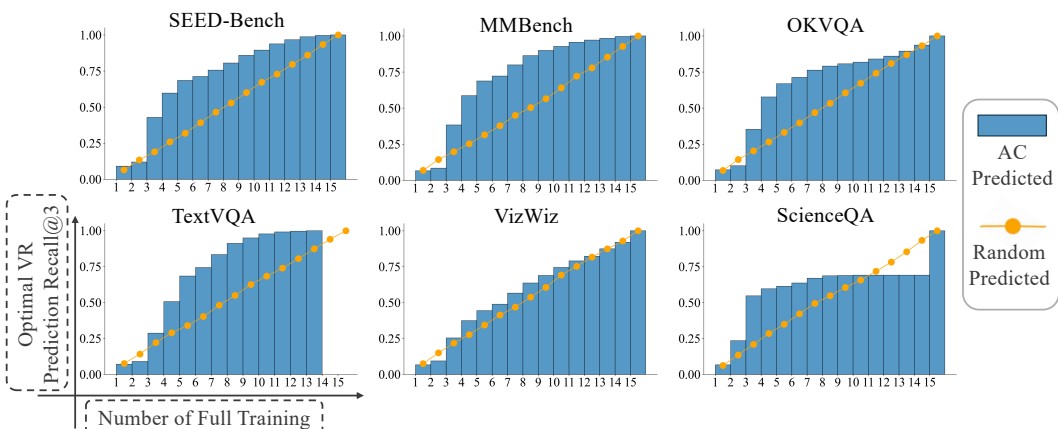

Figure 10: Number of full training (LLM finetuning) cycles required to include the optimal vision representation within the top-1 predictions (Recall@1).

- Reflects practical workflows—researchers typically inspect multiple ("top 3–5") candidates, not just one.
- Smooths score noise—unlike Top-1 accuracy, Recall@3 tolerates minor ranking fluctuations while still ensuring coverage of strong encoders.
- Focuses on coverage—rather than overall ranking correlation (e.g. Pearson's $r$) or mean reciprocal rank (which over-emphasizes exact positions), Recall@3 measures actionable retrieval quality: "Is the best encoder in my shortlist?" In other words, we put coverage in higher priority, "Is the best encoder in my shortlist" is more important than is the top-3 order correct.

While we believe this makes Recall@3 the most appropriate metric for AC's intended use-case, we also report Recall@1 for predicting the optimal vision representation. As shown in Figure 10, AC policy consistently outperforms random testing even under this stricter metric.

