# OpenReview forum: "Law of Vision Representation in MLLMs"
_colmweb.org/COLM/2025/Conference — COLM 2025_

### Official Review · Reviewer_ufKw · 2025-05-06

**Rating:** 6
**Confidence:** 4
**Ethics Flag:** 1

**Summary:**

This paper introduces the "Law of Vision Representation" for MLLMs, establishing a correlation between cross-modal alignment, visual representation correspondence, and model performance. Using quantitative A-score and C-score metrics, experiments across 15 visual representations and 8 benchmarks reveal a quadratic relationship between these scores and performance. The authors propose an AC policy to identify optimal visual representations without repeated language model fine-tuning, reducing computational costs by 99.7%.

**Reasons To Accept:**

1.This paper addresses an important challenge in the MLLM field—how to select optimal visual representations, which is currently often an empirical trial-and-error process consuming substantial computational resources. The writing is good, with a clearly defined and important problem statement.

2.The motivation is clear, with a relatively solid theoretical foundation. The authors not only propose a quantitative relationship between visual representation and model performance but also provide theoretical proofs in the appendix explaining why high alignment and high correspondence lead to better model performance.

3.The experimental design is relatively reasonable, covering 15 different visual representation configurations, including various encoder types, sizes, input resolutions, and feature combinations, while evaluating on 8 different types of benchmarks, including visual understanding and OCR tasks.

4.The work has practical value, as the proposed AC policy significantly reduces computational costs, which is important for both researchers and industrial applications.

**Reasons To Reject:**

1.The Law of Vision Representation specifically focuses on decoder-only MLLMs, which limits its application scope. How should other types of MLLMs be handled, and are there alternative approaches? For example, how should optimal visual representation models be selected for cross-attention-based MLLMs?

2.The paper proposes that A and C scores have a quadratic relationship with performance, but does not deeply explore why it's a quadratic relationship rather than another functional form. This point needs more in-depth analysis.

3.The paper does not sufficiently analyze the potential trade-off relationship between A and C, merely stating in Formula 3 that "AC Score to capture both the individual contributions of alignment and correspondence." What is the intrinsic connection between these two factors?

4.The paper experiments with different visual representation models. However, it only uses one base language model (Vicuna-7B 1.5) and does not verify applicability across language models of different scales and architectures (such as T5 model[1]).
[1] Raffel C, Shazeer N, Roberts A, et al. Exploring the limits of transfer learning with a unified text-to-text transformer[J]. Journal of machine learning research, 2020, 21(140): 1-67.

5.The radar chart on the right side of Figure 6 is rather blurry, making the text illegible.

---

> ### Author Response · Authors · 2025-06-03
> **Author Responses to Reviewer ufKw (1/3)**
>
> We appreciate your review of our work. Here is our responses to your comments:
>
> ---
>
> **Reviewer Comment:** *The Law of Vision Representation specifically focuses on decoder-only MLLMs, which limits its application scope. How should other types of MLLMs be handled, and are there alternative approaches? For example, how should optimal visual representation models be selected for cross-attention-based MLLMs?*
>
> **Author Response:** As noted in Section 3.1 of the paper, our “Law of Vision Representation” is deliberately framed around decoder-only MLLMs because these models currently dominate real-world practice and provide a clear, unified setting for comparing vision encoders. In contrast, cross-attention–based MLLMs (e.g., Q-former variants or Flamingo-style architectures) are not only far less popular but also do not share a common fusion mechanism. Each design uses its own cross-attention scheme, making it difficult to select a single framework that would allow us to derive a “law” for cross-attention MLLMs—much like how scaling laws for LLMs have been thoroughly characterized for decoder-only models, while encoder-only and encoder–decoder variants (e.g., BERT-style models and T5) remain under-explored in that context.
>
> Moreover, practical experience shows that cross-attention MLLMs typically require far more training data and model parameters to achieve reasonable vision–language alignment. Some examples include BLIP2 (129 M parameters) [1] and InstructBLIP [2], which use 15.9 M data points; Flamingo [3], which was trained on more than 2 B examples (2.1 B image–text pairs and 43 M interleaved data); and IDEFICS, which uses OBELICS (353 M images and 141 M English documents) [4].
>
> Attempting to extend our AC-score methodology to every individual cross-attention variant would require (1) defining a unified architectural template and (2) collecting large, specialized datasets to fine-tune and evaluate each design. Such efforts exceed the scope of the current manuscript. In that sense, a “Law of Vision Representation” for cross-attention MLLMs does not yet exist because the community itself has not converged on a single, scalable framework for these models.
>
> That said, we appreciate the reviewer’s suggestion to consider non–decoder-only MLLMs. We look forward to extending the current scope as soon as a unified, popular non–decoder-only MLLM emerges in our field.
>
> [1] Li, J., et al. (2023). BLIP-2: Bootstrapping Language-Image Pre-training with Frozen Image Encoders and Large Language Models. arXiv preprint arXiv:2301.12597.
>
> [2] Dai, W., et al. (2023). InstructBLIP: Towards Building Large Visual Instruction Models. arXiv preprint arXiv:2305.06500.
>
> [3] Alayrac, J.-B., et al. (2022). Flamingo: A Visual Language Model for Few-Shot Learning. arXiv preprint arXiv:2204.14198.
>
> [4] Laurencon, H., et al. (2023). OBELICS: An Open Web-Scale Filtered Dataset of Interleaved Image-Text Documents. arXiv preprint arXiv:2306.16527.

---

> ### Author Response · Authors · 2025-06-03
> **Author Responses to Reviewer ufKw (2/3)**
>
> **Reviewer Comment:** *The paper proposes that A and C scores have a quadratic relationship with performance, but does not deeply explore why it's a quadratic relationship rather than another functional form. This point needs more in-depth analysis.*
>
> **Author Response:** Here's our insight:
> - Good semantic correspondence means the vision part of MLLM can really tell what things are in a picture, even if they look a bit different. Like knowing all dogs are dogs, no matter their breed. But this alone isn't enough for the MLLM to "talk" about them. Those visual details need to be understood by the LLM.
> - Good language alignment means the visual information is perfectly understood by the language model. But if the initial visual information isn't very good at telling different things apart (poor semantic correspondence), then aligning it perfectly won't help much when the MLLM sees something new or slightly different.
>
> So, for MLLMs to work best, both semantic correspondence and language alignment need to work together. More importantly, they interplay with each other more than just adding up:
>
> If the visual features are already well-aligned with language (like in models such as CLIP), it actually helps the MLLM understand the semantic details better. That's because if you can map "dog" to the word "dog" consistently, it reinforces the concept of "dog" in the visual world.
>
> But why is this interplay represented in a quadratic form (like a hill)? Think of it in this way:
>
> 1. If language alignment is very weak, even if you make semantic correspondence a little better, the MLLM's performance won't jump much. It's like trying to have a good conversation when you don't speak the same language very well. Take a model like DINOv2. It's great at seeing things, but it might not be as good at linking those visual details to language as, say, CLIP. In this case, making the visual part even better won't help MLLM performance much because the language alignment is the bottleneck.
> 2. As language alignment improves, the MLLM can then really benefit from better semantic correspondence. It's like they're working together to reach that "sweet spot" where the performance really shines. Both getting better at the same time, or one unlocking the potential of the other, leads to this curved, quadratic rise in performance.
>
> In short, quadratic models include A² and C² and interaction term AC which includes exactly the relationship we want to capture. In our paper, we test linear, second-degree polynomial relationships. Here, we additionally test log-linear function fitting in the below table. Our result shows that the second-degree relationship fitted the best among others while still maintaining simplicity. More complex functional forms would obscure the clear individual contribution of A and C, as well as interaction between alignment and correspondence (which is discussed in the following question).
>
> | Fitting Method               | Vision R²  |
> |-----------------------------|------------|
> | Linear | 80.55 %    |
> | Quadratic   | 94.06 %    |
> | Log linear                  | 44.53 %    |

---

> ### Author Response · Authors · 2025-06-03
> **Author Responses to Reviewer ufKw (3/3)**
>
> **Reviewer Comment:** *The paper does not sufficiently analyze the potential trade-off relationship between A and C, merely stating in Formula 3 that "AC Score to capture both the individual contributions of alignment and correspondence." What is the intrinsic connection between these two factors?*
>
> **Author Response:** We thank the reviewer for pointing out the need to clarify the interaction between Alignment (A) and Correspondence (C). We address this point from two aspects: the ideal relationship between A and C suggested by the law, and the actual relationship we observe from existing encoders. To provide the ideal relationship, we extract the cross-term coefficient (α_AC) of the fitted functions. The table below shows α_AC for all eight benchmarks:
>
> | Benchmark        | α_AC     |
> |------------------|----------|
> | MMBench       | +1.1301  |
> | MME              | +1.5819  |
> | MMMU         | -0.4872  |
> | oOKVQA           | -0.0364  |
> | TextVQA      | +4.9804  |
> | VizWiz   | +2.9964  |
> | ScienceQA    | +4.4644  |
> | SeedBench       | +0.5772  |
>
> **Key findings:**
>
> 1. **Synergy (α_AC ≫ 0):** MMBench, MME, TextVQA, VizWiz, ScienceQA, and SeedBench all have α_AC > 0.
>    For these six tasks, performance improves **more than additively** when both A and C increase.
>
> 2. **Mild trade-off (α_AC < 0):** MMMU and OKVQA show α_AC < 0, indicating a slight penalty if A and C are both pushed upward (i.e., diminishing returns).
>
> These results demonstrate that on most benchmarks, Alignment and Correspondence actually reinforce each other (positive interaction). Therefore, A and C strengthen each other, so the ideal scenario would be increasing both to achieve the best performance.
>
> However, from our observations of A and C of vision encoders, the factors often have a trade-off: Intuitively, A measures how well image features align to a shared text embedding, whereas C captures intra-image consistency (e.g., visual cluster compactness). Forcing very high alignment often collapses visual cluster structure (hurting correspondence). In practice, looking at the A and C scores (provided in Appendix), common visual encoders often have high A but low C (CLIP and SigLIP) or low A but high C (DINOv2 and Diffusion features).
>
> ---
>
> **Reviewer Comment:** *The paper experiments with different visual representation models. However, it only uses one base language model (Vicuna-7B 1.5) and does not verify applicability across language models of different scales and architectures.*
>
> **Author Response:** Here, we additionally provide MLLM with Qwen2.5-14B-Instruct as LLM backbone fitting results. Specifically, we use the exact same settings (data, vision representations, hyperparameters), changing only the LLM backbone to Qwen 2.5 14B. We again evaluate the resulting MLLMs on 8 benchmarks, recompute the A and C scores, reporting the fitting R². The averaged fitting R² are above 90% for vision based benchmarks and above 80% for OCR based benchmarks, comparable to Vicuna 1.5 7B, showing that our conclusion applies to different types and sizes of LLM backbones.
>
> | Model         | Avg R² (Vision) | Avg R² (OCR) |
> |---------------|------------------|---------------|
> | Vicuna 1.5 7B | 94.06 %          | 83.85 %       |
> | Qwen 2.5 14B  | 91.50 %          | 82.27 %       |
>
> We appreciate the reviewer's insightful suggestion to explore T5-style MLLMs. However, the vast majority of current multimodal research, including leading models, focuses on decoder-only architectures. There isn't yet a widely accepted, unified framework or stable checkpoints for encoder-decoder T5-style MLLMs. Analyzing different T5-style MLLMs on a case-by-case basis would require redesigning each model individually, which may not guarantee generalizability.
>
> We believe our inclusion of Qwen2.5 14B already addresses the reviewer's concern about "different scales and architectures" within the prevailing decoder-only paradigm. We are confident that future work can explore encoder-decoder MLLMs once the field converges on a standard T5-fusion architecture and releases stable checkpoints. For this submission, we must humbly and respectfully decline to include T5 in this submission, but we genuinely appreciate the reviewer’s insightful suggestion.

---

> ### Author Response · Authors · 2025-06-08
> **Response to Reviewer ufKw**
>
> Thank you again for your thorough and insightful review. We hope our replies have addressed each of your concerns—especially regarding the decoder-only scope, the quadratic form justification, the AC interaction analysis, and the additional Qwen 2.5 14B results. If there is any point that remains unclear or any aspect you’d like us to expand upon further, please let us know and we will be happy to provide more details. We greatly appreciate your feedback.

---

> > ### Comment · Reviewer_ufKw · 2025-06-09
> >
> > Thank the authors you for the detailed explanations.
> > After reviewing other reviewers' comments,  I agree with them, I will increase my score.

---

> > > ### Author Response · Authors · 2025-06-09
> > >
> > > We sincerely thank you for revisiting our responses and raising the score! Your suggestions are really insightful and we will incorporate them in the final version.

---

### Official Review · Reviewer_UhUv · 2025-05-12

**Rating:** 6
**Confidence:** 3
**Ethics Flag:** 1

**Summary:**

This paper proposes the "Law of Vision Representation" in Multimodal Large Language Models (MLLMs), demonstrating a strong correlation between cross-modal alignment (A-score), representation correspondence (C-score), and overall model performance. Extensive experiments across diverse vision representations validate that model performance exhibits a quadratic correlation with the transformed function of A-score and C-score, which can be used to reduce computational costs when deciding the optimal vision representations for MLLMs.

**Questions To Authors:**

There are some unclear parts requiring further clarification:

About the Equation 3:
>  $\omega_{\alpha\beta}$ are trainable parameters optimized to best fit benchmark performance

For different downstream tasks, are these $\omega_{\alpha\beta}$ parameters learned from scratch, or updated through incremental learning? Is there a better way for cross-task adaption?

**Reasons To Accept:**

1. The authors propose a novel methodology utilizing visual-semantic alignment (A-score) and correspondence (C-score) metrics for optimal visual feature selection. This approach presents a theoretically sound and innovative solution.
2. The proposed method demonstrates significant computational efficiency improvements over traditional fine-tuning approaches, achieving substantial resource savings.
3. The experimental design is thorough and comprehensive, covering multiple vision representation. And the results are analyzed in depth.

**Reasons To Reject:**

1. In this paper, the authors propose a linear relationship between the AC score and model performance, termed the "Law of Vision Representation in MLLMs." However, have other potential factors been considered that might affect or weaken this conclusion? For instance:
 - Exploring a broader range of vision encoders
 - Investigating different types and scales of LLM backbones
 - Considering alternative visual representations or varying downstream tasks

2. Why is Recall@3 used as the evaluation indicator for the AC strategy, and what are its advantages compared to other indicators?

3. The illustration and writing can be improved for better understanding.

---

> ### Author Response · Authors · 2025-06-03
> **Author Responses to Reviewer UhUv (1/2)**
>
> We appreciate Reviewer UhUv for supporting our work, and we hope the responses below can make your decision more confident:
>
> ---
>
> **Reviewer Comment:** *In this paper, the authors propose a linear relationship between the AC score and model performance, termed the "Law of Vision Representation in MLLMs." However, have other potential factors been considered that might affect or weaken this conclusion?*
>
> **Author Response:**
>
> 1. *A broader range of vision encoders:* Our 15 encoders already cover a diverse set. Specifically, we covers two encoder architectures: feed-forward (SD 1.5, SD image variant) and transformer (CLIP, DINOv2); three encoder training mechanism: contrastive (CLIP, OpenCLIP, SigLIP, SigLIP2), self-supervised (DINOv2), diffusion (SD1.5, SDXL, DiT); multiple input resolution: CLIP@224, CLIP@336, SigLIP@256, etc; finally, we explore feature combinations of multiple encoders (CLIP + DINOv2). Additionally, we provide R^2 fitting on all 15 encoders and averaged R² fitting on subsets of 14 encoders, as shown in the below table. The small differences (< 1%) shows that showing no single one drives the Law, thus our selected vision representation is diverse enough.
>
> | Benchmark        | R² (no removal) | Avg R² (remove 1 encoder) |
> |------------------|------------------|----------------------------|
> | MMBench       | 95.06 %          | 95.47 %                    |
> | MME              | 95.35 %          | 95.73 %                    |
> | MMMU         | 70.70 %          | 71.43 %                    |
> | OKVQA           | 93.26 %          | 93.55 %                    |
> | TextVQA      | 95.73 %          | 95.85 %                    |
> | VizWiz   | 78.21 %          | 79.23 %                    |
> | ScienceQA    | 90.78 %          | 91.30 %                    |
> | SeedBench       | 92.58 %          | 93.08 %                    |
>
> 2. *Different types and scales of LLM backbones:* Here, we additionally provide MLLM with Qwen2.5-14B-Instruct as LLM backbone fitting results, as shown in the below table. Specifically, we use the exact same settings (data, vision representations, hyperparameters), changing only the LLM backbone to Qwen 2.5 14B. We again evaluate the resulting MLLMs on 8 benchmarks, recompute the A and C scores, reporting the fitting R². The averaged fitting R² are above 90% for vision based benchmarks and above 80% for OCR based benchmarks, comparable to Vicuna 1.5 7B, showing that our conclusion applies to different types and sizes of LLM backbones.
>
> | Model         | Avg R² (Vision) | Avg R² (OCR) |
> |---------------|------------------|---------------|
> | Vicuna 1.5 7B | 94.06 %          | 83.85 %       |
> | Qwen 2.5 14B  | 91.50 %          | 82.27 %       |
>
> 3. *Alternative visual representations or varying downstream tasks:* We have explored 15 visual representations on common ones like CLIP, OpenCLIP, SigLIP, and the recently released SigLIP2. Additionally, many of the vision representations have never been used on MLLM, such as diffusion features from SD 1.5, SD 2.1, SDXL, and DiT. The breadth of alternative visual representations already far exceeds what is used in mainstream MLLM works. The downstream tasks cover a diverse and popular set for evaluating MLLMs, including both vision (MMBench, MME, ScienceQA, SeedBench) and OCR based benchmarks (MMMU, VizWiz, TextVQA, OKVQA).

---

> > ### Comment · Reviewer_UhUv · 2025-06-07
> >
> > Thank the authors you for the detailed explanations.
> > I still keep positive about this paper.

---

> > > ### Author Response · Authors · 2025-06-08
> > > **Response to Reviewer UhUv**
> > >
> > > We sincerely appreciate your continued positive assessment of our work. Thank you for recognizing our work and for your constructive feedback throughout. We will incorporate your suggestions in the final version.

---

> ### Author Response · Authors · 2025-06-03
> **Author Responses to Reviewer UhUv (2/2)**
>
> **Reviewer Comment:** *Why is Recall@3 used as the evaluation indicator for the AC strategy, and what are its advantages compared to other indicators?*
>
> **Author Response:** Our AC policy’s core aim is to **narrow** the search space to a small set of promising vision encoders, rather than pinpointing a single best encoder. Recall@3 directly quantifies whether the true top-performers appear within our top-3 recommendations, which:
>
> 1. **Reflects practical workflows**—researchers typically inspect multiple (“top 3–5”) candidates, not just one.
> 2. **Smooths score noise**—unlike Top-1 accuracy, Recall@3 tolerates minor ranking fluctuations while still ensuring coverage of strong encoders.
> 3. **Focuses on coverage**—rather than overall ranking correlation (e.g. Pearson’s 𝑟) or mean reciprocal rank (which over-emphasizes exact positions), Recall@3 measures actionable retrieval quality: “Is the best encoder in my shortlist?” In other words, we put coverage in higher priority, “Is the best encoder in my shortlist” is more important than is the top-3 order correct.
>
> We believe this makes Recall@3 the most appropriate metric for AC’s intended use-case.
>
> ---
>
> **Reviewer Comment:** *For different downstream tasks, are these $\omega_{\alpha\beta}$ parameters learned from scratch, or updated through incremental learning? Is there a better way for cross-task adaption?*
>
> **Author Response:** To clarify our law fitting process: we fit a quadratic function of the form f(A, C) = β₀ + β₁A + β₂C + β₃A² + β₄(AC) + β₅C², where f(A, C) denotes benchmark performance, and A and C are the alignment and correspondence scores, respectively. The parameters ωₐᵦ (i.e., the β coefficients) are learned from scratch using ordinary least squares (OLS), yielding a closed-form solution. We use 15 data points (one for each vision encoder), which provides sufficient support for fitting the six parameters.
>
> Regarding cross-task adaptation, we believe the current approach is already efficient and practical. Since the model checkpoints are pre-obtained, computing evaluation scores and fitting the AC function for each benchmark requires minimal additional computation and introduces no extra cost.

---

### Official Review · Reviewer_Qc4i · 2025-05-13

**Rating:** 6
**Confidence:** 4
**Ethics Flag:** 1

**Summary:**

This paper investigates the relationship between vision representations and the performance of Multimodal Large Language Models (MLLMs). The authors propose the "Law of Vision Representation," suggesting that cross-modal alignment (A) and vision representation correspondence (C) are key determinants of MLLM performance. They introduce metrics to quantify A and C (A SCORE, C SCORE) and find a quadratic relationship between these scores and performance on various benchmarks. Based on this observed relationship, an "AC policy" is presented to efficiently select vision representations, aiming to reduce the extensive computational costs associated with empirically finetuning multiple MLLMs. Experiments are conducted with 15 vision representations and 8 benchmarks.

**Questions To Authors:**

See the Reasons To Reject

**Reasons To Accept:**

1. Empirical Exploration and Practical Utility:
The paper conducts a systematic empirical study across a considerable number of vision encoders and benchmarks, which is valuable for the community. The proposed AC policy, if robust, offers a tangible benefit by potentially reducing the computational resources needed to identify effective vision representations for MLLMs. The claimed 99.7% cost reduction is a significant practical motivation.

2. Quantification of Factors:
The effort to quantify concepts like cross-modal alignment and correspondence through specific metrics (A SCORE, C SCORE) is a step towards a more principled understanding, moving beyond purely heuristic selection of vision encoders.

**Reasons To Reject:**

1. Limited Conceptual Novelty and Overstated Claims ("Law"):
The primary concern with this paper is the perceived limited conceptual novelty. The core components used – the MLLM architecture (decoder-only), the vision encoders (CLIP, DINOv2, SigLIP, etc.), and the projector – are all well-established. The identified factors, cross-modal alignment and correspondence, are not new concepts in multimodal research. The contribution appears to be more in the systematic quantification and correlation analysis of these known factors within specific MLLM setups, rather than the discovery of fundamentally new principles or components.

2. Scope and Generalizability of the "Law" and AC Policy:
The definition of A SCORE (caption likelihood) and C SCORE (PCK on SPair-71k) might limit the "law's" applicability. As the authors rightly point out in their limitations, the C SCORE is not well-suited for OCR-heavy tasks. This suggests the current formulation of A and C might not be universally optimal predictive factors for all MLLM capabilities, weakening the claim of a general "law."

3. AC SCORE Formulation and Potential Overfitting:
The AC SCORE is derived by fitting a second-degree polynomial (with 6 trainable parameters $\tau v_{\alpha\beta}$) to benchmark performance using A and C scores from 15 vision representations. While the $R^2$ value is high (94.06%), fitting a 6-parameter model to 15 data points warrants caution regarding potential overfitting. The robustness of these $\tau v_{\alpha\beta}$ parameters across different sets of vision encoders or slight variations in benchmarks would be important to establish.

4. Practicality of A SCORE Calculation:
While the AC policy aims to save finetuning costs for the full LLM, the calculation of the A SCORE for *all* candidate vision encoders still requires Stage 1 training (training the projector). For a very large search space of vision representations, this initial step could still be computationally intensive.

---

> ### Author Response · Authors · 2025-06-03
> **Author Responses to Reviewer Qc4i (1/2)**
>
> We appreciate your review of our work. Here is our responses to your comments:
>
> ---
>
> **Reviewer Comment:** *Limited conceptual novelty and overstated claims ("Law")*
>
> **Author Response:** Our paper’s key novelty lies not in any single component (decoder‐only MLLM, CLIP/DINOv2 encoders, projector), but in the first-ever quantitative “Law of Vision Representation in MLLM” that links cross-modal alignment (A Score) and correspondence (C Score) to downstream performance via a simple quadratic model.
>
> Previous work on cross-modal alignment originates from the CLIP model [1, 2], where a contrastive objective naturally motivates investigation into alignment. In the context of MLLMs, X-VILA [3] mentions cross-modal alignment as a way to train across multiple modalities, such as image, text, and audio. Our work is the first in the MLLM field to formally introduce this concept and emphasize the importance of establishing strong cross-modal alignment. We provide both empirical and theoretical evidence showing that cross-modal alignment is directly correlated with MLLM benchmark performance.
>
> As for correspondence, the concept stems from traditional computer vision. To the best of our knowledge, no prior work in MLLMs has mentioned or quantified correspondence.
>
> **Our work is the first in MLLMs to jointly quantify cross-modal alignment and correspondence, demonstrate a quadratic relationship with downstream performance, and use this law to achieve substantial computational savings.** We demonstrate:
>
> 1. A quadratic “law” that predicts end-task performance with R² = 94.06% across 15 vision encoders and 8 benchmarks (Fig. 2);
> 2. A 99.7% reduction in vision representation searching cost using our AC policy (Section 4);
> 3. A theoretical justification for these findings, showing that well-aligned and high-correspondence representations both accelerate fine-tuning and enhance visual detail retrieval. Specifically, high A scores reduce the need for modality bridging during training, while high C scores enable more precise attention over image embeddings (Section 3.2).
>
> Additionally, we respectfully disagree with the assessment that our contribution merely involves systematic quantification and correlation analysis.
>
> Firstly, discovering new underlying laws by connecting previously known factors and revealing their inherent patterns is, in itself, a fundamental scientific contribution. For instance, foundational works like "Scaling Laws for Neural Language Models" [4] and "Training Compute-Optimal Large Language Models" [5] are celebrated precisely for deriving crucial insights by analyzing and correlating existing elements. Our paper similarly uncovers and quantifies novel, applicable relationships for MLLM training.
>
> Secondly, rigorous systematic quantification and correlation analysis are indispensable in LLM/MLLM research. This approach isn't a weakness; it's a vital methodology for understanding and optimizing complex systems. Dismissing such work as "not fundamentally new" unfairly mischaracterizes a necessary scientific approach.
>
> Finally, we systematically and theoretically identify and link these factors, leading to a simple and effective AC policy. This policy demonstrates clear, practical improvements in MLLM training efficiency, offering both new insights into MLLM dynamics and a concrete, effective optimization method.
>
> [1] Eslami, S., et al. (2023). Mitigate the Gap: Improving Cross-Modal Alignment in CLIP. arXiv preprint arXiv:2406.17639.
> [2] Gao, Y., et al. (2023). SoftCLIP: Softer Cross-Modal Alignment Makes CLIP Stronger. arXiv preprint arXiv:2303.17561.
> [3] Ye, H., et al. (2024). X-VILA: Cross-Modality Alignment for Large Language Model. arXiv preprint arXiv:2405.19335.
> [4] Kaplan, J., et al. (2020). Scaling Laws for Neural Language Models. arXiv preprint arXiv:2001.08361.
> [5] Hoffmann, J., et al. (2022). Training Compute-Optimal Large Language Models. arXiv preprint arXiv:2203.15556.
>
> ---
>
> **Reviewer Comment:** *Scope/generalizability limited by A/C definitions and OCR tasks*
>
> **Author Response:** We explicitly discuss this limitation in Section 6. However, the “law” remains predictive even in these settings: for example, our AC fit achieves R² = 83.85% on OCR benchmarks—well above random or single-factor baselines.
> To further improve applicability, a future step would be constructing a correspondence dataset specifically for OCR images. While this involves additional annotation, the overall framework of the law remains valid and adaptable.

---

> > ### Comment · Reviewer_Qc4i · 2025-06-08
> >
> > Thanks to the author's reply, my doubts have been basically resolved and I will increase my score.

---

> > > ### Author Response · Authors · 2025-06-08
> > > **Response to Reviewer Qc4i**
> > >
> > > We sincerely thank you for revisiting our responses and your positive feedback. We’re grateful for your thoughtful engagement and will incorporate your suggestions in the final version.

---

> ### Author Response · Authors · 2025-06-03
> **Author Responses to Reviewer Qc4i (2/2)**
>
> **Reviewer Comment:** *AC Score formulation and potential overfitting*
>
> **Author Response:** The quadratic function fitted on the AC scores effectively predicts the optimal set of vision representations, as demonstrated in Section 4. This application would not be successful if the relationship were merely due to overfitting.
>
> To demonstrate that our AC-based ranking is not overfit, we computed Spearman’s rank correlation (ρ) in a leave-one-out setting for all eight benchmarks. The table below shows both the in-sample (Full) and leave-one-out (LOOCV) ρ. In six benchmarks, LOOCV ρ remains above 0.73 (MMBench 0.882, MME 0.836, OKVQA 0.854, TextVQA 0.782, ScienceQA 0.729, SeedBench 0.879), indicating that, when any one encoder is held out, ≥ 73% of pairwise performance orderings are still correctly predicted. Only in two of the OCR-based benchmarks (VizWiz and MMMU), the fitted function generalizes less well (as discussed in Section 6), but this still indicates a significant monotonic trend.
>
> In summary, the small differences between Full vs. LOOCV ρ (average drop ≈ 0.084) across six benchmarks strongly support our claim that “AC score correlates with benchmark performance in a quadratic form,” and that this correlation generalizes robustly to unseen encoders rather than arising from overfitting.
>
> | Benchmark        | ρ (Full) | ρ (LOOCV) |
> |------------------|----------|-----------|
> | MMBench       | 0.957   | 0.882    |
> | MME              | 0.875   | 0.836    |
> | MMMU         | 0.800   | 0.676    |
> | OKVQA           | 0.943   | 0.854    |
> | TextVQA      | 0.832   | 0.782    |
> | VizWiz   | 0.839   | 0.464    |
> | ScienceQA    | 0.911   | 0.729    |
> | SeedBench       | 0.947   | 0.879    |
>
> ---
>
> **Reviewer Comment:** *Practicality—A Score still needs projector training for many encoders*
>
> **Author Response:** We politely point out that the effort of A score projector training is extremely minimal compared to full LLM training. Computing A Score (Stage 1) involves training only 0.298% of the model parameters on 558K image–caption pairs, requiring ≪1 GPU-day on a H100 GPU—significant less than full LLM finetuning. Moreover, projectors are reusable and can be cached; one can also prune large search spaces via a fast C Score pre‐filter before any A Score computation.

---

### Official Review · Reviewer_s8cX · 2025-05-19

**Rating:** 7
**Confidence:** 5
**Ethics Flag:** 1

**Summary:**

This paper presents a method of optimally selecting a frozen vision encoder for training multimodal large language models by using two metrics - cross-modal alignment (A) and vision representation correspondence (C). A refers to the alignment between the vision and text encoders and C refers to the ability of matching points in one image to their semantically corresponding points in other images. The theoretical justification presented for the correlation between high A and C scores and overall model performance is that high A score leads to less computation to bridge the gap between different modalities during training, and a high C score leads to precise attention in the image embeddings. The paper presents implementations of A score and C score and a combined AC score, verify the predictive power of the AC score and present a framework to sample vision representations from a set of vision encoders and given a language encoder and predict the optimal vision encoder using the AC score at a cost significantly less than finetuning all combinations to compare downstream performance. Finally, the paper presents experiments conducted with 15 vision encoders and 8 benchmarks. While the approach works on vision based benchmarks, its performance is limited on OCR-based benchmarks. The paper theorizes that this is due the C score being focused on natural images, and can be improved by adding OCR-based correspondence datasets.

**Reasons To Accept:**

+ Overall this is a comprehensive paper:
  + proposes a novel idea of using 2 metrics to predict downstream model performance.
  + provides a sound theoretical justification for the correlation between these scores and downstream performance
  + provides all the details required to understand the implementation of the scores
  + proposes a clear framework to use the scores in practice to select an optimal vision encoder
  + presents ample empirical validation of the idea with 15 vision encoders and 8 benchmarks
  + highlights the limitations of the approach and future directions.
+ The paper achieves a 99.7% reduction in the computational cost of selecting vision encoders and is able to predict downstream model performance using the A and C score with a coefficient of determination of 94.06%. This seems to be a promising result and could significantly simplify multimodal large language model training.

**Reasons To Reject:**

These are not strong reasons to reject.
+ The authors could include the computational cost of learning the AC scores and predicting the optimal vision encoder and contrast it with the traditional approach. The 99.7% reduction in computational cost is only mentioned in the abstract and not explained in the paper.
+ The method proposed in the paper is applicable only to multimodal models with a pretrained vision and language encoder. A future direction could be exploring the incorporation of A and C scores when learning the language and vision representations as part of model training.
+ As acknowledged by users, this approach works well only for natural image datasets and doesn’t do well for OCR datasets. Since these are an extremely important aspect of multimodal learning, performing well on these datasets would be pivotal for the applicability of this method.
+ It would be interesting to see how this approach scales to other modalities like audio.

---

> ### Author Response · Authors · 2025-06-03
> **Author Responses to Reviewer s8cX**
>
> We appreciate Reviewer s8cX for supporting our work. Here is our response to your insightful suggestions:
>
> ---
>
> **Reviewer Comment:**  *The authors could include the computational cost of learning the AC scores and predicting the optimal vision encoder and contrast it with the traditional approach. The 99.7% reduction in computational cost is only mentioned in the abstract and not explained in the paper.*
>
> **Author Response:** The 99.7% reduction in computational cost comes from the experiments that, after fitting the AC function, testing a new vision encoder only requires Stage 1 training (i.e., training a small projector), while traditional full MLLM post-training requires training both the projector and LLM.
>
>
> Concretely, we quantify computational cost using the number of trainable parameters, which is commonly correlated with both memory and compute. A decoder-only MLLM’s Stage 1 trains a small two-layer MLP, while Stage 2 involves fine-tuning a large language model with ∼7B parameters. This leads to a parameter ratio of about 0.003, yielding a 99.7% reduction.
>
> Moreover, if we use FLOPs as a more detailed metric (accounting for both model size and data scale), the savings become even more significant, since Stage 2 trains on more data and larger models than Stage 1.
>
> ---
>
> **Reviewer Comment:**  *The method proposed in the paper is applicable only to multimodal models with a pretrained vision and language encoder. A future direction could be exploring the incorporation of A and C scores when learning the language and vision representations as part of model training.*
>
> **Author Response:** We appreciate the reviewer’s insightful suggestion. Indeed, incorporating the A and C scores during joint training of vision and language representations is a promising direction. Unlike our current approach, where A and C are computed from fixed pretrained features, in a joint training scenario both the cross-modal alignment and correspondence would evolve dynamically throughout training.
>
> This introduces additional challenges, as both the vision and language representations co-adapt based on large-scale multimodal data, making it non-trivial to define A and C scores at any given point. One potential solution could be to compute A and C heuristics directly from the training data (e.g., evaluating how well each image-text pair is aligned or how semantically granular the visual features are).
>
> We consider this a valuable future direction and plan to explore it in future work. Thanks for the suggestion!
>
> ---
>
> **Reviewer Comment:** *As acknowledged by users, this approach works well only for natural image datasets and doesn’t do well for OCR datasets. Since these are an extremely important aspect of multimodal learning, performing well on these datasets would be pivotal for the applicability of this method.*
>
> **Author Response:** It is straightforward to extend our approach to OCR-based benchmarks. This would involve labeling a correspondence dataset specifically for OCR images. As this requires additional annotation effort and introduces a new layer of contribution, we plan to include it in our next round of work.

---

> > ### Comment · Reviewer_s8cX · 2025-06-09
> >
> > I thank the authors for their thoughtful responses. I will maintain my positive score of the paper. Thanks and good luck!

---

> ### Author Response · Authors · 2025-06-09
>
> We sincerely appreciate your continued positive support of our work again! We will incorporate your suggestions in the final version.

---

### Decision · Program_Chairs · 2025-07-08

**Decision:**

Accept

**Comment:**

This paper emphasizes the correlation between cross-modal alignment and correspondence scores (A and C scores) and MLLM performance. High A and C scores are indicative of superior model performance.
The authors further propose an AC policy for selecting optimal vision representations.
The AC policy selects the optimal vision representation for MLLMs, which significantly reduces computational costs and enhances both accuracy and efficiency in predicting the optimal vision representation

Pros
- The authors present comprehensive results on 15 vision encoders and 8 benchmarks.
- The AC policy has been demonstrated to efficiently predict the optimal vision representation, achieving a high prediction recall rate (e.g., 87.72% Recall@3) with minimal training resources, which is a significant improvement over random selection methods.
- The AC policy is versatile and can be applied to various types of vision representations.

Cons
- The applicability of the AC policy to other vision encoders and LLM architectures beyond those tested in the paper may be limited without further validation. The authors provide the additional results on Qwen2.5 and promise to include in the revised version.
- The paper acknowledges that the AC policy has a lower correlation with tasks that involve a significant amount of text in images (OCR-heavy), since the datasets used to compute the A and C scores do not adequately represent tasks that require understanding of tables, charts, and other text-heavy content.